# TARGETing secondary school students' motivation towards physical education: The role of student-perceived mastery climate teaching strategies

Gwen Weeldenburg[1,2]*, Lars Borghouts[1], Tim van de Laak[1], Teun Remmers[1], Menno Slingerland[1], Steven Vos[1,2]

1 School of Sport Studies, Fontys University of Applied Sciences, Eindhoven, The Netherlands,
2 Department of Industrial Design, Eindhoven University of Technology, Eindhoven, The Netherlands

* g.weeldenburg@fontys.nl

## Abstract

The aim of the present study was to explore the impact of TARGET-based teaching strategies on students' motivation in a Dutch secondary school PE context. We examined to what extent mastery climate teaching strategies perceived by students (independently or interactively) explain variability in students' motivation towards PE. In total 3,150 students (48.2% girls; 51.8% boys) with a mean age of 13.91 years (SD = 1.40) completed the Behavioural Regulations in Physical Education Questionnaire (BRPEQ), measuring students' autonomous motivation, controlled motivation and amotivation, and the Mastery Teaching Perception Questionnaire (MTP-Q), measuring student-perceived application of mastery TARGET teaching strategies. Hierarchical regression analyses indicated that after controlling for gender, age, and educational type, the predictive effects of the perceived mastery climate teaching strategies differed by motivational outcome. Overall, students who reported higher levels of perceived application of mastery TARGET teaching strategies showed more autonomous motivation and less amotivation. Specifically, the teaching strategies within the task structure were the strongest predictors for students' autonomous motivation and amotivation. No meaningful statistically significant two-way interaction effects between any of the TARGET variables were found, supporting the proposition of an additive relationship between the TARGET teaching strategies.

## Introduction

Positive experiences in physical education (PE) influence students' motivation and attitude towards physical activity (PA) and sport, which, in turn, increase the intention to participate in physical activities outside school [1–3]. In terms of promoting active involvement and effective learning within PE lessons themselves, the understanding of students' experiences and motivation can also be of great value for PE teachers. To this aim, the self-determination theory (SDT) [4, 5] can be helpful. This widely accepted and frequently applied theory of human motivation provides insight into what 'moves' students to action, or more specifically, what causes and energizes students' behaviour [5].

**Data Availability Statement:** All relevant data are within the manuscript and its Supporting Information files.

**Funding:** The author(s) received no specific funding for this work.

**Competing interests:** The authors have declared that no competing interests exist.

According to SDT, student motivation ranges from amotivation through controlled motivation (i.e., introjected and external regulation) to high-quality autonomous motivation (i.e., intrinsic motivation, integrated and identified regulation) [5]. Applied to a PE context, *amotivation* refers to a state in which students find no meaning or value in the PE activity and will likely have no intention to participate in it. *Controlled motivation* refers to students acting out of feelings of internal or external pressure, while *autonomous motivation* is characterized by students experiencing a sense of volition, identification, or enjoyment [5]. Overall, autonomous forms of student motivation are correlated with more adaptive outcomes, such as increased PA [6], enjoyment, concentration and vitality [2–4]. In contrast, controlled forms of student motivation and amotivation are associated with more maladaptive outcomes, such as disengagement, boredom and unhappiness [2–4]. According to SDT the different types of motivation are the result of social and environmental factors that can frustrate or satisfy students' basic psychological needs of autonomy (i.e., sense of volition and willingness), competence (i.e., experience of effectiveness and mastery) and relatedness (i.e., sense of connecting to and feeling important to others [4, 5]. By realizing a PE learning environment in which students' basic psychological needs are nurtured, the PE teacher is able to facilitate students' autonomous motivation [5–7], whereas a learning environment in which these psychological needs are frustrated results in controlled motivation and amotivation [5–7].

A contemporary theory detailing key aspects of the learning environment and related practical applicable teaching strategies, is the achievement goal theory (AGT) [8]. AGT provides an insight into how the environmental structure of, and situational cues from the teacher-created learning environment (i.e., motivational climate) affects student motivation [9]. Based on how students perceive the learning environment two types of motivational climates are defined: mastery (or task-involving) climate and performance (or ego-involving) climate [10, 11]. In a mastery climate the students perceive the importance of learning, personal development, effort towards task mastery, individual improvement, and collaboration (i.e., self-referenced criteria for success). In contrast, in a performance climate the students perceive that the emphasis is placed on results, winning and social comparison (i.e., norm-referenced criteria for success) [12]. Research in the context of PE and sport has demonstrated that mastery climates–in comparison to performance climates–are positively associated with a range of desirable learning and motivational outcomes [1, 9, 12].

Vansteenkiste et al. [13], have theorised that the goals of a behaviour as defined by AGT, can be considered the 'what' in motivation, whilst the underlying reasons for a behaviour as defined by SDT, constitute the 'why'. To gain enriching insights and a better understanding of potential underlying mechanism concerning students (lack of) motivation, scholars increasingly use this multi-theoretical perspective [14]. Several studies (e.g., [15, 16]) illustrated that the combination of the theoretical frameworks of the SDT and AGT can provide novel and practical insights. For example, García-Gonzáles et al. [17] showed that a mastery climate supports students' autonomous motivation through the experience of basic psychological needs satisfaction, while in a performance climate these needs are frustrated, leading to negative motivational outcomes.

Establishing a learning climate that supports student motivation and optimises learning and achievement is a complex task within the PE setting. Therefore, extensive knowledge of motivational climate structures and teaching strategies is needed [18]. In order to support PE teachers in creating a favourable mastery climate, the TARGET framework [19, 20] is considered useful [12, 21]. TARGET is an acronym for six teaching structures that enable the teacher to design effective teaching strategies to optimise the motivational PE learning climate: *task* (design of learning activities), *authority* (opportunities and location of decision-making), *recognition* (way of distribution of feedback and reinforcement), *grouping* (process and procedure

of grouping students), *evaluation* (a system or strategies for evaluating students learning) and *time* (pace of instruction and learning). A meta-analysis of 22 motivational climate intervention studies in PE employing the TARGET framework of Braithwaite et al. [12], showed adaptive outcomes for students experiencing a mastery climate and maladaptive outcomes for students experiencing a performance climate. More recent intervention studies confirmed this conclusion. [22] reported an increase in secondary school students' intrinsic motivation, identified regulation and introjected regulation, and a decrease in external regulation and amotivation when a mastery climate was created by applying TARGET-based teaching strategies in PE. In the study of [23] among Italian female secondary school students, the TARGET structures were manipulated to create either a mastery climate or a performance climate. The results showed that the mastery climate conditions correlated positively with pleasant/functional psychobiosocial (PBS) states, intrinsic motivation and identified regulation. In contrast, the performance climate conditions were positively related to unpleasant/dysfunctional PBS states, external regulations and amotivation.

Based on these studies, the TARGET framework can be considered valuable for teachers to identify, manage and manipulate dimensions within the learning climate that impact on student experiences and motivation within PE. However, to support PE teachers in positively impacting student motivation through the design of optimal learning environments, it is crucial to translate the TARGET structures into practical teaching strategies to establish a mastery climate, rather than a performance climate. Therefore, various authors have provided guidelines for effective teaching strategies based on the TARGET structures [22, 24–26]. Designing PE lessons based on these guidelines should theoretically enable teachers to realise a more mastery-oriented learning environment, which in turn should impact positively on student motivation. These guidelines, however, may not always be applicable in the same way, to every PE context. For example, strategies pertaining to student *authority* can be expected to be enacted differently in primary education compared to secondary education. Based on national PE standards, local school policies and guidelines, and school population, there is always a need for adaptation of these guidelines to the specific and unique context in which PE teachers find themselves. In line with this reasoning, previous research showed that the impact of TARGET teaching strategies on student perceptions of the motivational climate varies according to context [18, 27]. In order to effectively intervene and modify the learning environment, it is therefore suggested by several scholars [27, 28] to further examine the unique contribution of each TARGET structure within specific learning environments, and to identify which structures have the greatest impact on students' PE experiences and motivation.

Accordingly, the aim of the present study was to extend motivational climate research by exploring the impact of TARGET-based teaching strategies on students' motivation in a Dutch secondary school PE context. We examined to what extent the mastery climate teaching strategies perceived by students, independently or interactively correlate with and explain variability in students' motivation towards PE. Based on previous motivational climate research we hypothesised that higher levels of perceived mastery climate teaching strategies would relate positively to autonomous forms of motivation, whereas lower levels would primarily relate to controlled forms of motivation (i.e., introjected and external regulation) and amotivation.

## Methods

### Participants and procedures

Ethical approval was obtained by the Ethical Research Committee of Fontys University. For all participating students written informed parental/legal guardian consent and participant assent was obtained after they had received an information letter explaining the purpose of the study,

and its methods. In addition, permission to collect data with the students was obtained from the local school boards. The students were explained that the participation in the study was voluntary, and that there was guarantee of confidentiality and anonymity.

Participants were recruited by inviting PE teachers from a national PE network to participate in the study with their students. A convenience sample of 55 specialist PE teachers (22 females; 33 males; M age = 36.89; SD = 9.49) and their students from 12 secondary schools in the Netherlands participated in this cross-sectional study. In the Netherlands, all PE teachers in secondary education are specialist teachers who have obtained their teacher qualifications through a four-year physical education teacher education bachelor's course. PE in these schools was mixed gender grouped and mandatory for two lessons (of 50–60 minutes each) per week throughout the school year. In total 3,150 students (1518 girls; 48.2%, 1632 boys; 51.8%) with a mean age of 13.91 years (SD = 1.40) completed a web-based questionnaire near the end of the school year. The questions concerned students' PE experiences during the past schoolyear. The students took an average of 15 minutes to complete the questionnaire and were supervised by a teacher who was well informed about the procedure. The various educational tracks at the Dutch secondary school level were equally represented within the final sample: pre-vocational 36.1%; senior general 28.8%; and university preparatory 35.1%.

## Measures

**Student motivation.** Students' autonomous motivation, controlled motivation and amotivation towards PE in general was assessed using a modified version of the Behavioural Regulations in Physical Education Questionnaire (BRPEQ; [29]). The original BRPEQ includes 20 items reflecting autonomous motivation (i.e., intrinsic motivation, 4 items; identified regulation, 4 items), controlled motivation (i.e., introjected regulation, 4 items; external regulation, 4 items), and amotivation (4 items). BRPEQ, just as the BREQ-II it was based upon, does not measure integrated regulation, because research has shown that this is empirically indistinguishable from identified and intrinsic regulation through self-reports in children and adolescents [30]. To increase the usability and feasibility of the total questionnaire applied in this study, the number of items of the BRPEQ was reduced from 20 to 12 items based on the factor analyses by [30] and in line with previous research of [31]. The introductory stem 'In general I put effort in PE class. . .' was followed by 4 items reflecting autonomous motivation (2 x 2 items covering intrinsic motivation and identified regulation; e.g., 'because I enjoy it'; 'because I find PE personally meaningful'), 4 items reflected controlled motivation (2 x 2 items covering introjected and external regulation; e.g., 'because I have to prove myself'; 'because I feel the pressure of others to participate') and 4 items reflected amotivation (e.g., 'I find PE a waste of time'). All items were administered on a 5-point Likert scale ranging from 1 (strongly disagree) to 5 (strongly agree). A confirmatory factor analysis (CFA) with maximum-likelihood estimation was performed to test the factorial validity of the items extracting into autonomous motivation, controlled motivation and amotivation, by using Mplus 8.0 [32]. Prior to conducting the CFA, the sampling adequacy for analysis was tested by the Kaiser–Meyer–Olkin (KMO) measure. The KMO value was .89, which is well above the limit of .50 [33] and Bartlett's test of sphericity was significant (p < 0.001). Results of the CFA yielded an acceptable fit $\chi^2$ = 539.459, p < .001, CFI = .971, RMSEA = .061, SRMR = .067. Internal consistencies for autonomous motivation and amotivation were satisfactory with Cronbach's alphas of .87 and .85, respectively. The Cronbach's alpha of .66 for controlled motivation was less satisfactory. However, [34] notes that when dealing with psychological constructs, values below .70 can, realistically, be expected. Furthermore, a lower number of items can have a profound negative effect on alpha [35]. Since there were only four items for controlled motivation, and deleting items

from the analysis had no relevant positive effect on the Cronbach's alpha, we decided to retain the controlled motivation scale for further analysis. Similar to previous research [29, 31] we calculated composite scores of autonomous motivation, controlled motivation, and amotivation.

**Student-perceived application of mastery climate teaching strategies.** To assess to what extent different mastery climate teaching strategies were applied by the teacher during PE according to students, we employed a newly constructed questionnaire, the Mastery Teaching Perception Questionnaire (MTP-Q). The initial items were generated by the authors of this study and based on the guidelines and suggestions of the TARGET framework literature [10, 11, 22, 24–26]. The MTP-Q started with 33-items representing teaching strategies within the TARGET framework structures. Since the Time structure is considered inextricably linked to the task and evaluation structure [10], and relatively abstract to operationalize independently, we chose not to query the time structure as a separate structure in this study. After the items were pilot tested in a target age group (N = 8) for feasibility and comprehensibility, they were subjected to an exploratory factor analysis (EFA) using SPSS version 26.0. A principal component method with Promax for oblique rotation (because factors were assumed to be correlated) was used to explore the underlying theoretical structure (i.e., task, authority, recognition, grouping and evaluation structure). Prior, the KMO measure verified the sample adequacy for the analysis with a value of .96 and a significant Bartlett's test of sphericity (p < 0.001). The EFA yielded five factors with an eigenvalue above 1, explaining 57,47% of the variance in the mastery climate teaching strategy items. After rotation, these factors could be interpreted as representing the task, authority, recognition, grouping and evaluation structure. All items loaded with interpretable factor loadings with absolute values greater than .40 [36]. Items that cross-loaded substantially with other factors were first critically and independently reviewed on face and content validity by four experts in the fields of didactics and motivation in PE, and subsequently discussed. As a result, seven items were deleted and three items were moved to a different TARGET construct. The questionnaire was reduced to 26 items (see S1 Questionnaire) representing the motivational mastery climate structures of *task* (6 items; e.g., 'There is plenty of variety and alternation in the PE lessons'), *authority* (5 items; e.g., 'Our PE teacher provides opportunities for all students to deliver input and ideas during PE lessons'), *recognition* (5 items; e.g., 'Our PE teacher provides me with personal attention and feedback as much as possible'), *grouping* (5 items; e.g., 'During the grouping process our PE teacher ensures that all students feel equally valued and that no-one is excluded'), and *evaluation* (5 items; e.g., 'During assessment within PE, student differences (e.g., height, weight, strength) are taken into account'). After an introduction in which it was explained that the purpose of the questionnaire was to gain insight into how the student experienced PE in general last school year, students rated each item on a 5-point Likert scale ranging from 1 (strongly disagree) to 5 (strongly agree). The internal consistency reliability coefficients ($\alpha$) for these subscales were satisfactory with Cronbach's alphas of .79, .79, .85, .80, and .82 for *task*, *authority*, *recognition*, *grouping* and *evaluation*, respectively. We also created a composite mastery teaching strategy score by averaging the respective scores of the *task*, *authority*, *recognition*, *grouping* and *evaluation* structure to test our hypothesis that higher levels of perceived mastery climate teaching strategies would relate positively to autonomous forms of motivation and lower levels primarily relate to controlled forms of motivation and amotivation.

**Data analysis.** Preliminary analyses consisted of descriptive analyses (means and standard deviations) and the calculation of Pearson's bivariate correlations to explore the relationships between the perceived mastery climate teaching strategies and motivation types of students. We explored whether adjusting for the multilevel nested structure of our data (students nested within teachers and teachers within schools) would lead to improvements in the quality of our

models and in the estimates of individual parameters. First, one-way ANOVAs revealed that variation between the 12 schools was not statistically significant (F = 0.86, F = 0.85, and F = 0.85 for autonomous motivation, controlled motivation, and amotivation, respectively). Also, variations between the 55 teachers were not statistically significant (F = 1.12, F = 1.14 and F = 0.90 for autonomous motivation, controlled motivation, and amotivation, respectively). Second, we explored whether adding between-school variance as a fixed factor would increase the quality of the model, but for all motivational constructs this was not the case (p >.05). Third, we compared the quality of intercept-only models without- and with the addition of a random intercept for schools and for teachers. Results showed that for both schools and teachers, differences between both models were not statistically significant (i.e., -2 log likelihood difference-score between models ranged between 0.12 and 1.37; p >.05). Therefore, we decided to conduct ordinary least squares multivariate regression analysis using SPSS version 26.0. After examining the assumptions of normality, linearity, homoscedasticity and absence of multicollinearity [34], we first conducted a series of three regression analyses (for each of the motivational outcomes) to determine the combined influences of the teaching strategies within all TARGET structures (i.e., mastery teaching strategy composite score) on students' motivation. Subsequently, three series of hierarchical regression were performed to investigate the extent to which each of the five TARGET structure could predict the motivation of students. As gender, age, and educational type are known to potentially affect students' motivation [37, 38] these variables were entered in the first step of each regression analysis to control for their effects. In the second step of this regression, the student-perceived mastery climate teaching strategies were simultaneously entered to examine whether they could account for additional variance of students' motivation. We analysed the influence of each TARGET structure while controlling for all other TARGET structures to reveal independent influences on students' motivation. Finally, we also explored whether individual TARGET structures interacted with each other in predicting students' motivation. Therefore, we computed two-way interaction terms between the TARGET variables.

## Results

### Descriptive statistics and correlational analysis

Means, standard deviations, internal consistency reliability coefficients, and bivariate correlations for all study variables are presented in Table 1. Student motivation for PE is more autonomously regulated (3.56) rather than controlled (2.09), with amotivation showing similar scores to controlled motivation (2.15). Regarding the student-perceived application of mastery climate teaching strategies, the students reported the highest mean scores on recognition (3.66) and task (3.56) strategies and the lowest mean score on authority (3.00). For the grouping and evaluation strategies mean scores above the scale midpoint, of respectively 3.37 and 3.44 were reported. Positive and significant correlations (Table 1) between perceived mastery climate teaching strategies and autonomous motivation were found. In contrast, amotivation was significantly negatively correlated with perceived mastery climate teaching strategies. Correlations between the teaching strategies and controlled motivation were either low or negligible.

### Regression analyses

To test our hypothesis in which we assumed that higher levels of overall student-perceived mastery climate teaching strategies would relate positively to autonomous forms of motivation, and negatively to controlled forms of motivation and amotivation, we conducted linear regressions based on the composite mastery teaching strategy score (M = 3.40; SD = .65). For

**Table 1. Means, standard deviations, Cronbach's alpha, and correlations of all study variables.**

| Variable | M | SD | Scale | α | 1. | 2. | 3. | 4. | 5. | 6. | 7. |
|----------|---|----|----|---|----|----|----|----|----|----|----|
| 1. Autonomous motivation | 3.56 | .98 | 1–5 | .87 | - | | | | | | |
| 2. Controlled motivation | 2.09 | .78 | 1–5 | .66 | .03 | - | | | | | |
| 3. Amotivation | 2.15 | 1.04 | 1–5 | .85 | -.66* | .26* | - | | | | |
| 4. Task | 3.56 | .70 | 1–5 | .79 | .50* | .01 | -.35* | - | | | |
| 5. Authority | 3.00 | .85 | 1–5 | .79 | .38* | .14* | -.17* | .54* | - | | |
| 6. Recognition | 3.66 | .79 | 1–5 | .85 | .43* | .00 | -.31* | .66* | .55* | - | |
| 7. Grouping | 3.37 | .80 | 1–5 | .80 | .38* | .06* | -.24* | .57* | .58* | .64* | - |
| 8. Evaluation | 3.44 | .79 | 1–5 | .82 | .44* | .02 | -.30* | .61* | .56* | .69* | .63* |

* Correlation is significant at the 0.01 level (2-tailed).

*autonomous motivation* a significant regression equation was found $F(5, 3144) = 224.605$, $p < .001$), with a $R^2$ of .26. Higher levels of perceived mastery teaching strategies positively influenced students' autonomous motivation ($B = .776$, $p < .001$). For *controlled motivation* a significant regression equation was found ($F(5, 3144) = 2.658$, $p = .021$), with a $R^2$ of .004, however the impact of the perceived mastery teaching strategies on *controlled motivation* was negligible ($B = .070$, $p = .001$). Regarding *amotivation* a significant regression equation was found $F(5, 3144) = 80.559$, $p < .001$), with a $R^2$ of .11. Higher levels of perceived mastery teaching strategies decrease students' amotivation ($B = -.526$, $p < .001$).

With respect to the influence of individual TARGET structures on *autonomous motivation* (see Table 2) we found that the overall model was significant $F(9, 3140) = 98.055$, $p < .001$,

**Table 2. Summary of hierarchical regression analysis for variables predicting autonomous motivation.**

| Variables | B | SE | β | R² | ΔR² |
|-----------|---|----|----|----|----|
| *Step 1* | | | | .00 | |
| Gender (boys = 1, girls = 2) | .02 | .04 | .01 | | |
| Age | .01 | .01 | .02 | | |
| Educational Type 1 | ref. | ref. | ref. | | |
| Educational Type 2 | .09 | .04 | .04* | | |
| Educational Type 3 | .03 | .04 | .02 | | |
| *Step 2* | | | | .29 | .29 |
| Gender (boys = 1, girls = 2) | .02 | .03 | .01 | | |
| Age | .00 | .01 | .00 | | |
| Educational Type 1 | ref. | ref. | ref. | | |
| Educational Type 2 | .08 | .04 | .04* | | |
| Educational Type 3 | .00 | .04 | .00 | | |
| Task | .45 | .03 | .32** | | |
| Authority | .09 | .02 | .08** | | |
| Recognition | .09 | .03 | .07** | | |
| Grouping | .02 | .03 | .02 | | |
| Evaluation | .18 | .03 | .14** | | |

*Note.* N = 3150; *B* = Unstandardised regression coefficient; *SE* = Standard error; *β* = Standardised regression coefficient; *R²* = Amount of variance explained; Educational Type 1 = pre-vocational, Educational Type 2 = senior general, Educational Type 3 = university preparatory

*p < .05.

**p < .01.

$R^2 = .29$ (see Table 2). The results showed that the inclusion of students' gender, age, and educational type in Step 1 did not significantly contribute to the prediction of autonomous motivation ($R^2 = .00$). After controlling for these variables, autonomous motivation was significantly predicted by higher student-perceived application of the task ($\beta = .32$), evaluation ($\beta = .14$). authority ($\beta = .08$) and recognition ($\beta = .07$) teaching strategies. The level of student-perceived grouping strategies ($\beta = .02$) was not a significant predictor for autonomous motivation.

The overall model predicting controlled motivation (Table 3) was also significant, $F(9, 3140) = 10.679$, $p < .001$, $R2 = .03$ (see Table 3). The analysis revealed that Step 1 did not explain any variance in the outcome variable ($R2 = .00$). After controlling for gender, age, and educational type, controlled motivation was significantly predicted by higher levels of student-perceived application of the authority ($\beta = .20$), and grouping teaching strategies ($\beta = .05$), and lower perceived application levels of the recognition strategies ($\beta = -.09$). The task and evaluation strategies were not significant predictors.

The overall model predicting *amotivation* (Table 4) was significant, $F(9, 3140) = 62.220$, $p < .001$, $R^2 = .15$ (see Table 4). The inclusion of students' gender, age, and educational type in Step 1 predicted a significant amount of variance ($R^2 = .01$), with gender, and educational type as significant predictors. The addition of the mastery climate teaching strategies in Step 2 improved the model predicting students' amotivation. This dependent variable was significantly predicted by lower levels of student-perceived task ($\beta = -.26$), evaluation ($\beta = -.13$), and recognition ($\beta = -.10$) teaching strategies, and higher levels of the authority teaching strategies ($\beta = .10$). The level of student-perceived application of grouping strategies ($\beta = .00$) was not a significant predictor.

**Table 3. Summary of hierarchical regression analysis for variables predicting controlled motivation.**

| Variables | B | SE | β | $R^2$ | $\Delta R^2$ |
|---|---|---|---|---|---|
| *Step 1* | | | | .00 | |
| Gender (boys = 1, girls = 2) | -.01 | .03 | -.01 | | |
| Age | .00 | .01 | .00 | | |
| Educational Type 1 | *ref.* | *ref.* | *ref.* | | |
| Educational Type 2 | -.02 | .04 | -.01 | | |
| Educational Type 3 | .03 | .03 | .02 | | |
| *Step 2* | | | | .03 | .03 |
| Gender (boys = 1, girls = 2) | -.01 | .03 | -.01 | | |
| Age | .00 | .01 | .00 | | |
| Educational Type 1 | *ref.* | *ref.* | *ref.* | | |
| Educational Type 2 | -.01 | .03 | -.01 | | |
| Educational Type 3 | .03 | .03 | .02 | | |
| Task | -.05 | .03 | -.04 | | |
| Authority | .18 | .02 | .20** | | |
| Recognition | -.09 | .03 | -.09** | | |
| Grouping | .05 | .02 | .05* | | |
| Evaluation | -.04 | .03 | -.04 | | |

*Note.* N = 3150; *B* = Unstandardised regression coefficient; *SE* = Standard error; *β* = Standardised regression coefficient; *$R^2$* = Amount of variance explained; Educational Type 1 = pre-vocational, Educational Type 2 = senior general, Educational Type 3 = university preparatory

*p < .05.

**p < .01.

**Table 4. Summary of hierarchical regression analysis for variables predicting amotivation.**

| Variables | B | SE | β | $R^2$ | $\Delta R^2$ |
|---|---|---|---|---|---|
| *Step 1* | | | | .01 | |
| Gender (boys = 1, girls = 2) | -.07 | .04 | -.04* | | |
| Age | .00 | .01 | .00 | | |
| Educational Type 1 | *ref.* | *ref.* | *ref.* | | |
| Educational Type 2 | -.18 | .05 | -.08** | | |
| Educational Type 3 | .01 | .04 | .00 | | |
| *Step 2* | | | | .15 | .14 |
| Gender (boys = 1, girls = 2) | -.07 | .03 | -.04* | | |
| Age | .00 | .01 | .00 | | |
| Educational Type 1 | *ref.* | *ref.* | *ref.* | | |
| Educational Type 2 | -.16 | .04 | -.07** | | |
| Educational Type 3 | .02 | .04 | .01 | | |
| Task | -.38 | .03 | -.26** | | |
| Authority | .13 | .03 | .10** | | |
| Recognition | -.13 | .03 | -.10** | | |
| Grouping | .00 | .03 | .00 | | |
| Evaluation | -.18 | .03 | -.13** | | |

*Note*. N = 3150; *B* = Unstandardised regression coefficient; *SE* = Standard error; *β* = Standardised regression coefficient; *$R^2$* = Amount of variance explained; Educational Type 1 = pre-vocational, Educational Type 2 = senior general, Educational Type 3 = university preparatory

*p < .05.

**p < .01.

No evidence of statistically significant two-way interaction effects between the predictor variables was found for autonomous motivation and amotivation (p > .05 for all interactions). With regard to controlled motivation a significant two-way interaction effect was found for task × recognition (F(23, 3126) = 5.368), p < .001). Given the negative interaction coefficient (*B* = -.123, p = .003) these predictors seem to have a combined inhibitory effect on controlled motivation.

## Discussion and conclusion

The purpose of this study was to gain more insight into how PE teachers can effectively manipulate the teaching environment to foster a mastery climate and with that potentially affect student motivation positively. Therefore, we examined the impact of the student-perceived application of mastery climate teaching strategies on secondary school students' motivation. As recommended by several scholars [27, 28] we also examined the unique contribution of the individual TARGET structures to identify which structures have the greatest impact on students' motivation in the context of PE. Consistent with our hypothesis, students who reported higher levels of perceived application of mastery task, authority, recognition, grouping and evaluation teaching strategies showed higher levels of autonomous motivation and less amotivation. These findings are in line with AGT theory and empirical studies which have shown that perceptions of a higher mastery-orientated climate are associated with autonomous motivational regulations [9, 17, 23]. However, the results of the hierarchical regression analyses provided us with a more detailed understanding and revealed that the predictive effects of the perceived mastery climate teaching strategies differed by motivational outcome.

The teaching strategies within the *task* structure, as defined in our questionnaire, were the strongest positive predictor for autonomous motivation and the strongest negative predictor for amotivation. Consistent with the work by [39–41] these findings highlight the importance of providing alternation and variation, challenging yet achievable tasks, sufficient opportunities for involvement and PA, and brief and clear instructions concerning the learning task for students' motivation within PE. [42] argue that the recognition and evaluation structures are the most decisive for students' perceptions of the motivational climate. However, their study did not actually statistically test the inter-relationship between the predictor (i.e., TARGET strategies) and outcome variables (i.e., perceived motivational climate). In contrast, the results of the present study indicate that the *task* structure is the strongest predictor of students' motivation. The rationale for this could be that the learning tasks form the backbone of a PE lesson, to which the other TARGET structures are connected, and by which they are valued and become meaningful. Contrary to, for example, the strategies within the grouping structure, the students will almost constantly be confronted with the characteristics of the learning task, possibly explaining why this impacts students' experiences the most.

The perceived strategies within the *evaluation* structure also positively predicted autonomous motivation and negatively predicted amotivation, albeit less strong than the task structure. In line with TARGET literature and the teaching strategies outlined in our MTP-Q, PE teachers should focus on individual development and growth rather than on normative results, set reasonable and achievable goals and take the differences between students into account. More specifically related to assessment, teachers are advised to create transparency concerning learning goals and assessment criteria (feed-up), to select valid and reliable assessment tools, and to provide students with sufficient information about their current performance (feedback) and how they can improve (feedforward). These findings reinforce prior research which has shown that evaluation or assessment activities, can play a pivotal role in students' motivation. For example, [43] found that transparency about the assessment criteria positively impacts students' motivation. In a recent qualitative study [26], secondary school students voiced their preference for an emphasis on the individual learning process and -progress during PE evaluation/assessment, rather than on normative learning outcomes.

As part of autonomy-supportive teaching behaviour, it has been suggested to facilitate and stimulate students' involvement in the decision-making processes and have them assume leadership roles within PE as part of autonomy-supportive teaching behaviour [2, 5]. Our findings partially support this notion. However, the teaching strategies within the *authority* structure were also associated with controlled forms of motivation and amotivation. These results may suggest that autonomy-supportive strategies such as providing students with choice, responsibility and opportunities to deliver input and ideas during PE, do not affect the motivation of all students in the same way. Indeed [44] demonstrated that students with different motivational profiles benefited differently from need-supportive teaching strategies. Some students might feel empowered when, for example, they are actively involved in the evaluation or grouping process, while other, perhaps less competent students might experience pressure. On the other hand, studies such as [45] have suggested that autonomy-supportive teaching is beneficial to all students, independent of their motivational regulations. These apparently conflicting views may be due to the complexity of motivational teaching styles and their impact on students' perceptions, as pointed out by [6]. Using a circumplex model, they provided a refined insight into the various dimensions of (de)motivating teaching styles (i.e., autonomy support, control, structure and chaos) and how these relate to each other. The model showed that the motivating autonomy-supportive style, and in particular its subarea termed the 'participative approach' (e.g., encouraging student initiative, offering choices) is adjacent to the demotivating 'chaotic' teaching style. In this latter style, clear communication of expectations, guidelines

for student behaviour, and tailored guidance and support is missing or insufficient, resulting in a perceived lack of structure by students [6]. Several studies (e.g., [46, 47]) have demonstrated that teacher structure is imperative for more adaptive motivational outcomes. Yet, on the other side of the spectrum, teachers should avoid (over) structuring and becoming too directive, since this could result in a demotivating, controlling teaching style. Taken together, this implies that within the authority structure of TARGET, the PE teacher should be constantly aiming for a functional balance of supporting students' autonomy and providing ample structure.

The TARGET structures are considered to function interdependently, which argues for an integrative approach while teaching [10]. However, there is still debate on how these structures interact, and whether they operate in an additive or a multiplicative manner. If the structures are additive, they become complementary and can compensate for each other. In other words, students can still perceive the learning environment as motivating (i.e., mastery climate), despite inadequacies in some TARGET structures. Deficiencies in for example the evaluation structure could then be attenuated by strengths in the recognition structure. However, if these structures would work in a multiplicative manner, this compensation mechanism does not apply, and all TARGET structures would need to be mastery-focused for students to perceive an overall motivating climate. The work of [18, 42] suggested the existence of an additive relationship between the TARGET teaching structures. These suggestions, however, were primarily based on logical reasoning rather than statistical analyses. The absence of meaningful, significant interaction effects between the TARGET variables in the present study now lends statistical support to the additive relationship proposition. Based on this insight, teachers could focus on the implementation of mastery climate teaching strategies within selected TARGET structures, rather than focusing on all TARGET structures at the same time. Especially for unexperienced teachers this could reduce the complexity of PE teaching practice.

The mastery TARGET teaching strategies seem to have little to no impact on controlled motivation, in contrast to our findings on autonomous motivation and amotivation. Apparently, the levels of mastery teaching strategies do not seem to influence students' feelings of external pressure (i.e., external motivational regulation) and internal pressure (i.e., internal motivational regulation). Although this result is in line with the study of [17], who found no relation between the motivational climate and external regulation, we cannot provide a logical or research-based explanation for this finding. We therefore support the call of García-González and colleagues [17] that there is a need for studies that investigate the impact of the TARGET teaching strategies on the different motivational subtypes distinguished in SDT.

## Strengths and limitations

To our knowledge, this is the first study in which the TARGET framework was operationalised into a validated questionnaire, to examine how perceived teaching strategies within the TARGET structures independently or interactively impact on secondary school students' motivation. We believe our findings provide support for, and a detailed insight into the additive mechanistic nature of the TARGET structures. However, this study is not without limitations. First, the application of the TARGET teaching strategies was determined by the perceptions of students only. Therefore, we have no objective data on the actual application of the teaching strategies. Second, although we found statistically significant results, the moderate amount of explained variance in our models suggests that there are other variables affecting students' motivation within PE, in addition to the perceived TARGET teaching strategies. The present study does not allow to determine the nature of these variables. Third, by including students through a convenience sample of volunteering PE teachers from our university network, the

PE teachers may not have been representative for the entire population of secondary PE teachers. In theory, this might also have impacted upon student-perceived teaching strategies.

## Conclusions

The application of mastery TARGET teaching strategies within secondary school PE has a positive impact on students' motivation. Due to the additive relationship of the TARGET structures, teachers could focus on the implementation of mastery climate teaching strategies within selected TARGET structures, rather than focusing on all TARGET structures at the same time. In that case it is recommended to put emphasis on the application of the mastery teaching strategies within the *task* structure because this seems to be the most decisive factor for student motivation.

## Supporting information

**S1 Dataset.**
(SAV)

**S1 Questionnaire. Mastery Teaching Perception Questionnaire (MTP-Q).**
(DOCX)

## Acknowledgments

We want to thank Luuk van Iperen for providing us assistance with the data analysis, and all PE teachers and students that participated in this study.

## Author Contributions

**Supervision:** Steven Vos.

**Writing – original draft:** Gwen Weeldenburg.

**Writing – review & editing:** Lars Borghouts, Tim van de Laak, Teun Remmers, Menno Slingerland, Steven Vos.

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
