## [Decision Letter · Decision Letter 0]

23 Jun 2022

PONE-D-22-02985TARGETing secondary school students’ motivation towards physical education: The role of student-perceived mastery climate teaching strategies.PLOS ONE

Dear Dr. Weeldenburg,

Thank you for submitting your manuscript to PLOS ONE. After careful consideration, we feel that it has merit but does not fully meet PLOS ONE’s publication criteria as it currently stands. Therefore, we invite you to submit a revised version of the manuscript that addresses the points raised during the review process.

We look forward to receiving your revised manuscript.

Kind regards,

Francisco Javier Huertas-Delgado, Ph.D.

Academic Editor

PLOS ONE

Journal Requirements:

Reviewers' comments:

Reviewer's Responses to Questions

**Comments to the Author**

1. Is the manuscript technically sound, and do the data support the conclusions?

Reviewer #1: Yes

Reviewer #2: Partly

2. Has the statistical analysis been performed appropriately and rigorously? 

Reviewer #1: I Don't Know

Reviewer #2: Yes

3. Have the authors made all data underlying the findings in their manuscript fully available?

Reviewer #1: Yes

Reviewer #2: Yes

4. Is the manuscript presented in an intelligible fashion and written in standard English?

Reviewer #1: Yes

Reviewer #2: Yes

5. Review Comments to the Author

Reviewer #1: This research may be of interest to the readers of the journal. Authors present an interesting study about the impact of TARGET-based teaching strategies on students’ motivation in a Dutch secondary school PE context. The importance of the topic is indubitable. However, as the authors point out in the section on limitations, the application of the TARGET teaching strategies was determined by the perceptions of students only. Therefore, they have no objective data on the actual application of the teaching strategies.

According to the authors, this is the first study in which the TARGET framework was operationalised into a validated questionnaire, to examine how perceived teaching strategies within the TARGET structures independently or interactively impact on secondary school students’ motivation. This study can be an interesting starting point to explore this topic further.

In relation to the text, please, see some comments and recommendations below:

Introduction:

It is a manuscript, not a chapter, so I recommend to reduce and summarize the information of this section. It is important to note the significant number of references incorporated in the introduction, although they are not very recent. The structure of the introduction follows the submission guidelines. The aim of the research is clear and relevant.

Methods:

The research design seems to fit the purpose of the research. The instruments used for data collection are well described.

Results

The results section is well structured and very detailed.

Discussion and conclusions:

It is important to note the significant number of references incorporated in the discussion, although they are not very recent.

The conclusions are presented clearly and are justified by the results presented.

References:

Barely 25% of the references are later than 2016. It is necessary to include more up-to-date bibliography. If it is a topic of interest, it should be evidenced in the knowledge generated in recent years or, at least, in the discrepancies and suggestions of other authors, on the importance of addressing research on this topic.

The format of references should be revised to comply with the submission guidelines (https://journals.plos.org/plosone/s/submission-guidelines).

Reviewer #2: The manuscript is extremely interesting and well written. However, it would be necessary some reviews in the manuscript before been publish. There are two topics that need to be clear informed.

[page 8, line 165; 171-174] Ethics statement should be written in a specific topic in methods section. And more details about the institution where the project was approved.

[page 8, line 167] Why a convenience sample was selected instead of a probabilistic sample? Wasn’t this an additional limitation of the study? Also, it was not clear the eligibility criteria for specialist PE teachers being selected for the study. Would be interesting for readers to have some information about the universe population of specialist PE teachers in Netherlands.

6. PLOS authors have the option to publish the peer review history of their article (what does this mean?). If published, this will include your full peer review and any attached files.

Reviewer #1: No

Reviewer #2: **Yes: **Jessyka Mary Vasconcelos Barbosa

---

## [Author Response · Author response to Decision Letter 0]

11 Jul 2022

Dear dr. Francisco Javier Huertas-Delgado,

Thank you for the opportunity to revise and resubmit our manuscript. Below, we will address all comments made by you and the reviewers.

Editor’s comments:

>> We have changed the file names accordingly and have attempted to meet all PLOS ONE style requirements. This included, for example, changing header font sizes, adapting table headers and changing the in-text references from round to square brackets.

>> We have removed the phrase ‘data not shown’. We meant to indicate that the data was only shown in-text, not in a table.

>> We have changed this accordingly.

>> We have thoroughly reformatted all references, including addition of doi’s to all papers. As far as we are aware, there is now no citation of retracted papers.

Reviewers' comments:

Reviewer #1: 

Introduction:

It is a manuscript, not a chapter, so I recommend to reduce and summarize the information of this section. It is important to note the significant number of references incorporated in the introduction, although they are not very recent. The structure of the introduction follows the submission guidelines. The aim of the research is clear and relevant.

>> We have abbreviated this section by removing some of the more ‘textbook’-like information, e.g. elaborate information about SDT-motivational regulations, as well as the practical examples of lessons strategies associated with TARGET structures. We agree that these do not necessarily add to the understanding of our research objectives.

Also, we have deleted a number of references that were basically ‘overlapping’, making sure to retain the more recent ones where possible (see later).

Methods:

The research design seems to fit the purpose of the research. The instruments used for data collection are well described.

Results

The results section is well structured and very detailed.

Discussion and conclusions:

It is important to note the significant number of references incorporated in the discussion, although they are not very recent.

The conclusions are presented clearly and are justified by the results presented.

References:

Barely 25% of the references are later than 2016. It is necessary to include more up-to-date bibliography. If it is a topic of interest, it should be evidenced in the knowledge generated in recent years or, at least, in the discrepancies and suggestions of other authors, on the importance of addressing research on this topic.\\

>> We have added a number of more recent references, so that now 49% of all references is post-2016.

The format of references should be revised to comply with the submission guidelines (https://journals.plos.org/plosone/s/submission-guidelines).

>> We have changed the format of the references to comply with the journal’s guidelines.

Reviewer #2: The manuscript is extremely interesting and well written. However, it would be necessary some reviews in the manuscript before been publish. There are two topics that need to be clear informed.

[page 8, line 165; 171-174] Ethics statement should be written in a specific topic in methods section. And more details about the institution where the project was approved.

>> We have added more info about the ethical procedures and approval and combined this into one paragraph (183-188).

[page 8, line 167] Why a convenience sample was selected instead of a probabilistic sample? Wasn’t this an additional limitation of the study? Also, it was not clear the eligibility criteria for specialist PE teachers being selected for the study. Would be interesting for readers to have some information about the universe population of specialist PE teachers in Netherlands.

>> We added text to the limitation to acknowledge this (501-505). Also, we included some information about (specialist) PE teachers in secondary education in the Netherlands (192-195).

---

## [Decision Letter · Decision Letter 1]

8 Sep 2022

TARGETing secondary school students’ motivation towards physical education: The role of student-perceived mastery climate teaching strategies.

PONE-D-22-02985R1

Dear Dr. Weeldenburg,

We’re pleased to inform you that your manuscript has been judged scientifically suitable for publication and will be formally accepted for publication once it meets all outstanding technical requirements.

Kind regards,

Gwo-Jen Hwang

Academic Editor

PLOS ONE

Additional Editor Comments (optional):

Reviewers' comments:

Reviewer's Responses to Questions

**Comments to the Author**

1. If the authors have adequately addressed your comments raised in a previous round of review and you feel that this manuscript is now acceptable for publication, you may indicate that here to bypass the “Comments to the Author” section, enter your conflict of interest statement in the “Confidential to Editor” section, and submit your "Accept" recommendation.

Reviewer #1: All comments have been addressed

2. Is the manuscript technically sound, and do the data support the conclusions?

Reviewer #1: Yes

3. Has the statistical analysis been performed appropriately and rigorously? 

Reviewer #1: I Don't Know

4. Have the authors made all data underlying the findings in their manuscript fully available?

Reviewer #1: Yes

5. Is the manuscript presented in an intelligible fashion and written in standard English?

Reviewer #1: Yes

6. Review Comments to the Author

Reviewer #1: The response of the authors of the work have been satisfactory. The detailed explanations as well as the specification of the new version of the manuscript are appreciated. The submitted work may be of great interest to the readers of the journal.

7. PLOS authors have the option to publish the peer review history of their article (what does this mean?). If published, this will include your full peer review and any attached files.

Reviewer #1: **Yes: **Javier Rico-Díaz

---

## [Editor Report · Acceptance letter]

13 Sep 2022

PONE-D-22-02985R1 

TARGETing secondary school students’ motivation towards physical education: The role of student-perceived mastery climate teaching strategies. 

Dear Dr. Weeldenburg:

I'm pleased to inform you that your manuscript has been deemed suitable for publication in PLOS ONE. Congratulations! Your manuscript is now with our production department. 

Kind regards, 

on behalf of

Dr. Gwo-Jen Hwang 

Academic Editor

PLOS ONE